# Chemical-Free Biologically Enhanced Primary Treatment of Raw Wastewater for Improved Capture Carbon

Muhammad Rizwan Azhar [1,*], Paul Nolan [2], Keith Cadee [2] and Mehdi Khiadani [1]

[1]  School of Engineering, Edith Cowan University, 270 Joondalup Drive, Joondalup, WA 6025, Australia
[2]  Water Corporation, 629 Newcastle Street, Leederville, WA 6007, Australia
*  Correspondence: m.azhar@ecu.edu.au; Tel.: +61-863045877

**Abstract:** Conventional wastewater treatment processes require extensive energy inputs for their operations. Biologically enhanced primary treatment (BEPT) is a promising technology to capture incoming organics that may be utilized to produce biogas and potentially hydrogen with further downstream processing. This study involved a biologically enhanced primary treatment (BEPT) of raw wastewater at bench and pilot-scale using activated sludge (AS) addition and dissolved air flotation (DAF) using raw wastewater at a municipal wastewater facility in Western Australia with average chemical oxygen demand of ~800 mg/L. The results of pilot-scale testing showed an improved removal performance for total chemical oxygen demand (COD-T), soluble chemical oxygen demand (COD-S), and total suspended solids (TSS) compared to conventional primary treatment (PT). Specifically, average COD-T, COD-S and TSS removals for BEPT were 33.3%, 13.5% and 45%, respectively which was 10%, 100% and 6% higher than PT. Moreover, the sludge produced from BEPT had a high solids content of 4.8 g/L, which might not need further thickening prior to anaerobic digestion. It is important to note that no chemicals were used during BEPT testing, which makes the process very cost-effective.

**Keywords:** chemical oxygen demand; dissolved air flotation; pilot scale; enhanced primary treatment; wastewater

## 1. Introduction

Rapid population increases and urbanization have led to an increase in wastewater generation; thus, sustainable wastewater treatment has gained importance in recent years. Normally, wastewater treatment processes require extensive energy inputs to remove unwanted materials, i.e., water pollutants, mainly organic materials and nutrients, e.g., ammonium, nitrates and phosphorous [1]. Due to the stringent environmental regulations, climate change and depletion of fossil fuels, there is a need for more sustainable wastewater treatment processes [2]. This will not only deal with wastewater treatment, but also include water reuse and the recovery of energy and resources, e.g., nutrients. Similarly, the dependence on nutrients from natural resources, e.g., phosphorus rocks for fertilizers, is a driving force to look for a more sustainable process for phosphorous recovery from wastewater treatment plants [3]. To reduce energy requirements and increase sustainability, wastewater treatment plants (WWTPs) need to be transformed into water resource recovery facilities (WRRFs) where several products, such as water, energy in different forms (heat, electricity, fuel, etc.), bioplastics, and nutrients, are produced. To achieve energy neutrality in WRRFs, innovative treatment technologies need to be developed, which require lower energy inputs for carbon and nutrient removal [4,5]. This goal can be achieved by retrofitting these new technologies into existing WWTPs [6,7]. Despite some remarkable achievements in resource recovery in recent last decades at small scale, most WWTP operations are still far from reaching this goal [8].

Although WWTPs have been in use for more than a century [9], most plants use variants of the conventional activated sludge (CAS) process. In the CAS process, incoming





organic materials, usually quantified as chemical oxygen demand (COD), are converted into carbon dioxide, water, and biomass. This is achieved by supplying oxygen/air to degrade these large organic molecules with the help of microorganisms [10]. The energy requirements in WWTPs largely consist of energy for aeration, which is dependent on the quality (COD and nutrients) of incoming wastewater. Aeration requirements for CAS (average COD 300–800 mg/L) require 30–60% of the total energy requirements of a WWTP [11,12]. Conversely, the COD can be captured as energy-rich solids, which can be converted to energy with the residual used in agriculture. Many existing WWTPs utilize anaerobic digestion to convert captured COD into biogas (methane), which may then be used in energy production. These processes are best-suited to medium and large plants [13,14].

Wastewater is a rich source of energy. Inherently, there are two types of energy in wastewater, chemical and thermal energy. Chemical energy can be sub-divided into two categories, i.e., the carbonaceous matter, which is normally quantified by the COD, (3.49 Wh/g COD or 12.6 kJ/g COD) and inorganic ammonia, but this is difficult to exploit [15,16]. In most situations, the COD concentration is a good indicator of the energy recovery potential of wastewater, with the COD concentration of medium-strength municipal wastewaters typically being between 400 and 500 mg/L [17,18]. The other form of energy that can be recovered in primary treatment is thermal energy [19,20]. The potential energy recovery from municipal wastewater globally is around 70–140 GW of continuous energy supply, equivalent to 52–104 million tons of oil annually [21].

Therefore, harvesting the chemical energy (COD) from municipal wastewater and then converting it into biogas through anaerobic digesters and/or incinerator is a practical means of energy recovery. The advanced primary treatment of raw wastewater, such as biologically enhanced primary treatment and chemically enhanced primary treatment (CEPT), can serve as newer ways of recovering organic matter such as COD. In advanced primary treatment, organic carbon can be captured upfront, hence reducing the load on secondary treatment and recovering organic carbon to potentially produce energy. There are various alternatives to conventional primary treatment, such as high-rated activated sludge (HRAS), chemically enhanced primary treatment (CEPT) or membrane bioreactors (MBR) to concentrate the organic matter present in the wastewater [22–27]. HRAS systems have been extensively studied as an alternative to primary treatment, as has CEPT. It should be noted that although CEPT has been shown to perform well compared to conventional primary treatment, this improved performance comes with the additional costs of coagulants and flocculants and is a more complicated process. For example, Rabaey et al. studied HRAS-DAF systems with the addition of a single and binary polymer in addition to iron chloride or aluminium chloride. Higher removal efficiencies were reported for COD-T, COD-S and TSS using cationic/anionic polymers and metal salts [28].

Here, we describe the integration of activated sludge (AS) addition and DAF processes for the chemical-free primary treatment of wastewater similar to the Captivator® process. The Captivator® process is an advanced form of primary treatment, which replaces conventional clarification and sludge-thickening with contact stabilization with aeration in the presence of waste-activated sludge, followed by DAF for sludge separation. Although, the Captivator® process has been implemented at a large scale with and without the addition of chemicals at a limited number of sites, the variability in wastewater characteristics requires site-specific investigations. Moreover, implementation of the Captivator® process depends on the plant layout, raw wastewater characteristics and available sludge solids [29,30]. In this study, bench-scale and pilot-scale testing were carried out using real municipal wastewater at Subiaco Water Resource Recovery Facility (SWRRF) in Perth, Western Australia. Promising results have been reported based on COD-T, COD-S and TSS removals with removal average removal efficiencies of 33.3%, 13.5% and 45%.

## 2. Experimental

### 2.1. Operational Setup for BEPT System

There were three main components of the BEPT pilot plant, i.e., contact tank, sludge tank and DAF tank. Prior to the pilot plant study, bench-scale experiments were performed to find the optimum operating parameters using a Platypus DAF jar tester and Lovibond's floc unit with four 1.5 L jars (Figure S1). In BEPT pilot plant, the contact tank, activated sludge was mixed with raw wastewater to achieve the biological uptake of organics from the raw wastewater. Aeration was provided by a small aeration pump system and air stones, with the supply of air adjusted by inserting/removing the air stones in the contact tank. No chemical flocculants were used in this study, only activated sludge and raw wastewater.

In the DAF unit, air-saturated water was introduced to generate micro-bubbles to separate suspended solids through a floatation process; a representative image of bubbles is shown in Figure S2.

### 2.2. Biologically Enhanced Primary Treatment (BEPT) Experiments

Raw wastewater (after screening and grit removal) was supplied from Subiaco water resource recovery facilities (SWRRFs) Perth, through a transfer pump, while sludge was collected in a 25-L tank from the returned activated sludge (RAS) pipe at the plant. The solid contents in RAS were determined by collecting 50 mL RAS into a vial. Then, this was filtered using a vacuum filtration unit using Whatman filter paper 1. The retained solids were dried in a microwave oven. The quantity of solids was obtained by subtracting mass of empty filter paper from mass of filter with dried solid. In normal DAF rig operation, raw wastewater with total chemical oxygen demand (COD) ~800 mg/L was supplied to feed tank of DAF and the flow was controlled with a control valve to introduce wastewater into contact tank. The contact tank was continuously mixed and aerated by means of fine-bubble aeration at a dissolved oxygen (DO) set point between 1.0 and 1.5 mg L, monitored with a DO meter. Activated sludge (AS) was introduced to the contact tank by a peristaltic pump at a specific flow rate. A schematic representation of the pilot-scale BEPT process is presented in Figure 1. Compressed air is supplied to the DAF saturator at a pressure of 5.5 bar to the DAF unit recycle stream. This so-called "saturated white water" was de-pressurized into the contact zone of the DAF unit through three nozzles, which caused microbubbles to form. The sludge float from the DAF unit was removed by a skimmer, and collected in a non-aerated tank with an approximate retention time of 1.5 h. The testing period extended to ~200 days with natural variation in wastewater composition and temperature. The COD-T, COD-S, and TSS, analyses were performed according to Standard Methods for Examination of Water and Wastewater [31]. Specifically, COD-T was measured by collecting the samples into 50 mL vial, with 2 mL samples extracted from the vial with a pipette and introduced to COD measuring kit (K-7365, Range: 0–1500 ppm (HR) USEPA). For COD-S measurements, 50 mL samples were filtered through a vacuum filtration unit, then 2 mL samples were introduced into COD measuring kit. TSS was measured by drying filtered solids onto the filter paper (Whatman filter paper 1).

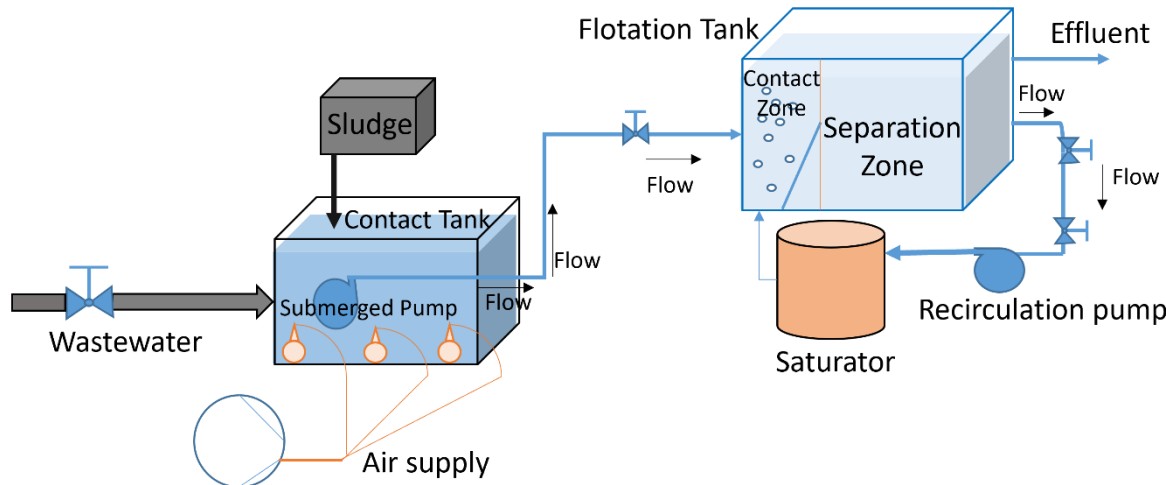

**Figure 1.** Schematic representation of experimental setup for the pilot-scale BEPT process.

### 3. Results and Discussions

#### 3.1. Characteristics of Influent and Return Activated Sludge

A portion of raw wastewater from the main wastewater treatment was divrted to the pilot-scale experimental set up, which was used directly for BEPT testing. The COD-T, TSS and ranged between 700 and 1100 mg/L and 350 and 550 mg/L, with a pH 7.6–8.2, respectively. Moreover, RAS has total solids in the range from 5400 to 6500 mg/L, with average values of 5800 mg/L. The average concentration of nutrients as total kjeldahl nitrogen (TKN) was ~75 mg/L.

#### 3.2. Bench Scale Testing

To study the effectiveness of DAF, initial bench top experiments were conducted without adding any activated sludge. It was observed that more than 50% of total suspended solids were removed in these experiments. The results are consistent with the reported literature [28]. Therefore, further tests were conducted using low sludge loadings, i.e., 6.25, 12.5 and 25 mL/L (denoted as SL1, SL2 and SL3, respectively) and the results are presented in Figure S3. Generally, no significant difference was observed in COD removal efficiency even at lower AS loadings. It is a well-established fact that DAF technology can effectively remove suspended solids [32,33]; however, for BEPT, dissolved COD capture, along with total suspend solids (TSS), are required to recover organics for energy production.

Further tests were conducted using low AS loadings with varying flow rates of air saturated water to study their effect on COD removal efficiency. It is important to mention that flow rate was roughly estimated using the time of introducing 300–600 mL of air saturated deionized (DI) water into contact jars. At lower flow rates of air-saturated DI water, there was no significant separation of TSS, and only a very thin float formed after DAF treatment, probably due to there being an insufficient number of bubbles in the jar. However, with the introduction of more air-saturated DI water into contact tanks, a thick floc was formed at the surface and considerable reductions in COD-T, COD-S and TSS were observed. Lower removal rates with a higher sludge loading for SL3 experiments might be a result of mixing solids from float and/or reduced distribution of bubbles in the jar. Lower removal rates can also be attributed to larger flocs, which are not separated by smaller bubbles [34].

For the bench-scale testing, less accurate results were expected due to the limitations of the jar testing equipment. Further testing was conducted on a pilot-scale rig to more accurately study the effectiveness of DAF in BEPT performance in continuous operation.

### 3.3. Pilot Scale Results

A detailed analysis of COD-T, COD-S and TSS removal was conducted for around 200 days (June 2020 to May 2021) during the steady, continuous operation of the BEPT process. The results were also compared with the historical results of Subiaco WRF for the previous two years (2019–2021). It can be seen from Table 1 that the pilot-scale BEPT process outperformed the existing PT process at the SWRRF in terms of average COD-T, COD-S and TSS removals at much shorter detention times. Moreover, the high solid concentration in the float may be directly used in an anaerobic digester for biogas production. The performance of the pilot-scale BEPT process is shown in Figure 2.

**Table 1.** Performance comparison of pilot-scale BEPT process and conventional primary treatment of wastewater.

|  | Maximum | Minimum | Average | Std. Deviation |
|---|---|---|---|---|
| Biological Enhanced Primary Treatment with DAF | | | | |
| COD-T Removal% | 48.2 | 18.1 | 33.3 | 6.4 |
| COD-S Removal% | 36 | 2.3 | 13.5 | 6.6 |
| TSS Removal% | 78.8 | 22.6 | 44.6 | 8 |
| Primary Sedimentation Tank | | | | |
| Subiaco PT COD-T Removal% | 40.3 | 3.6 | 23.9 | 9.6 |
| COD-S Removal% | N.A | N.A | N.A | N.A |
| TSS Removal% | 70.4 | 0.23 | 38.6 | 12.2 |

Notes: COD-T Is Total Chemical Oxygen Demand and COD-S Is Soluble Chemical Oxygen Demand Performance Index.

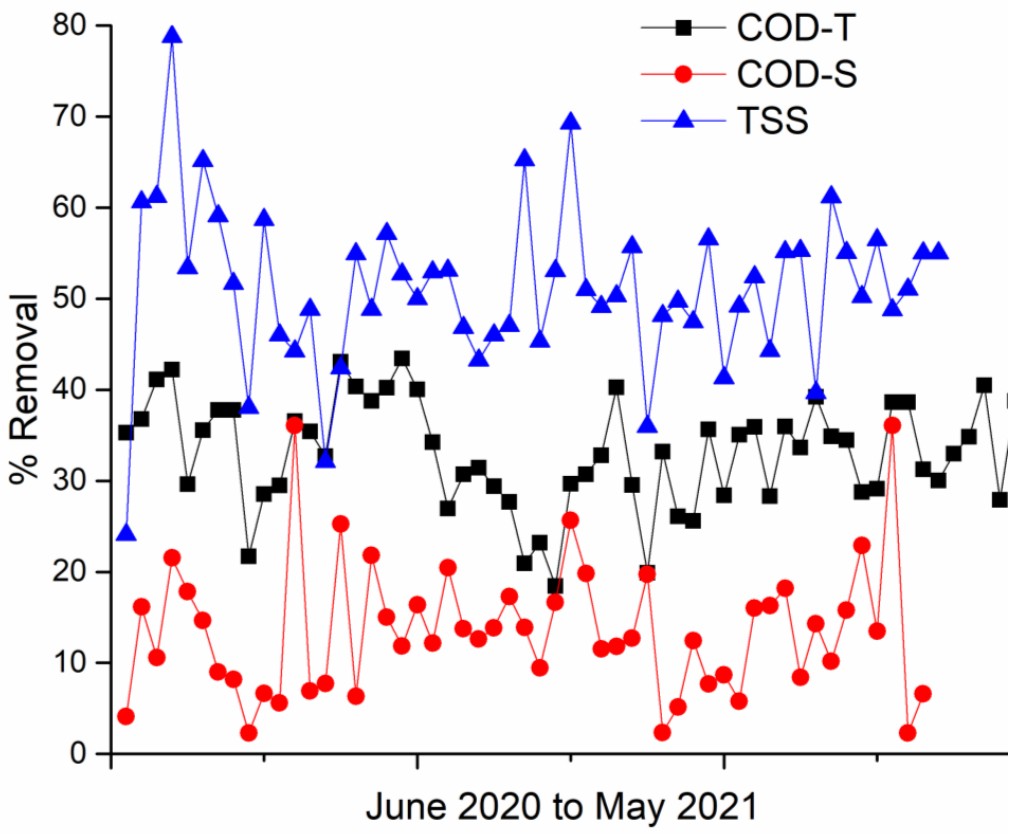

**Figure 2.** Removal efficiency of COD-T, COD-S and TSS in pilot-scale BEPT process.

### 3.4. Effect of Hydraulic Retention Time

There was a significant improvement in the performance of the pilot-scale BEPT process when the HRT in the contact tank increased from 15 min to 30 min, but there was only a marginal improvement in COD-T, COD-S and TSS removal with a further increase in HRT to 45 min. It can be seen from Figure 3 that a 10% increase in COD-T was observed for HRT of 30 min compared to HRT of 15 min, while no further improvement was observed when the HRT was increased from 30 and 45 min. The trend was similar for COD-S, where approximately 6% more soluble COD was removed when doubling HRT from 15 to 30 min. On the other hand, TSS removal was very low in the case of 15 min HRT compared to HRTs of 30 and 45 min. A lower HRTs in the contact tank also reduce the degree of mineralization and maximize the organics redirected to the sludge via biological flocculation, sorption, and storage mechanisms [10,14]. Higher flow rates effectively increased the hydraulic loading rate (HLR) in DAF tank, which may have had a negative effect on floc formation due to turbulence in the DAF tank in the absence of any chemical dosing. The hydrodynamics of a flotation, including bubble rise velocity, is the key factor deciding the collision and attachment efficiency of solid particles. This affects the overall separation efficiency of a micro-bubble flotation system. If the rise velocity is too high, it will result in high turbulence and can break bubble-floc particles, which could result in low separation efficiency [33,35].

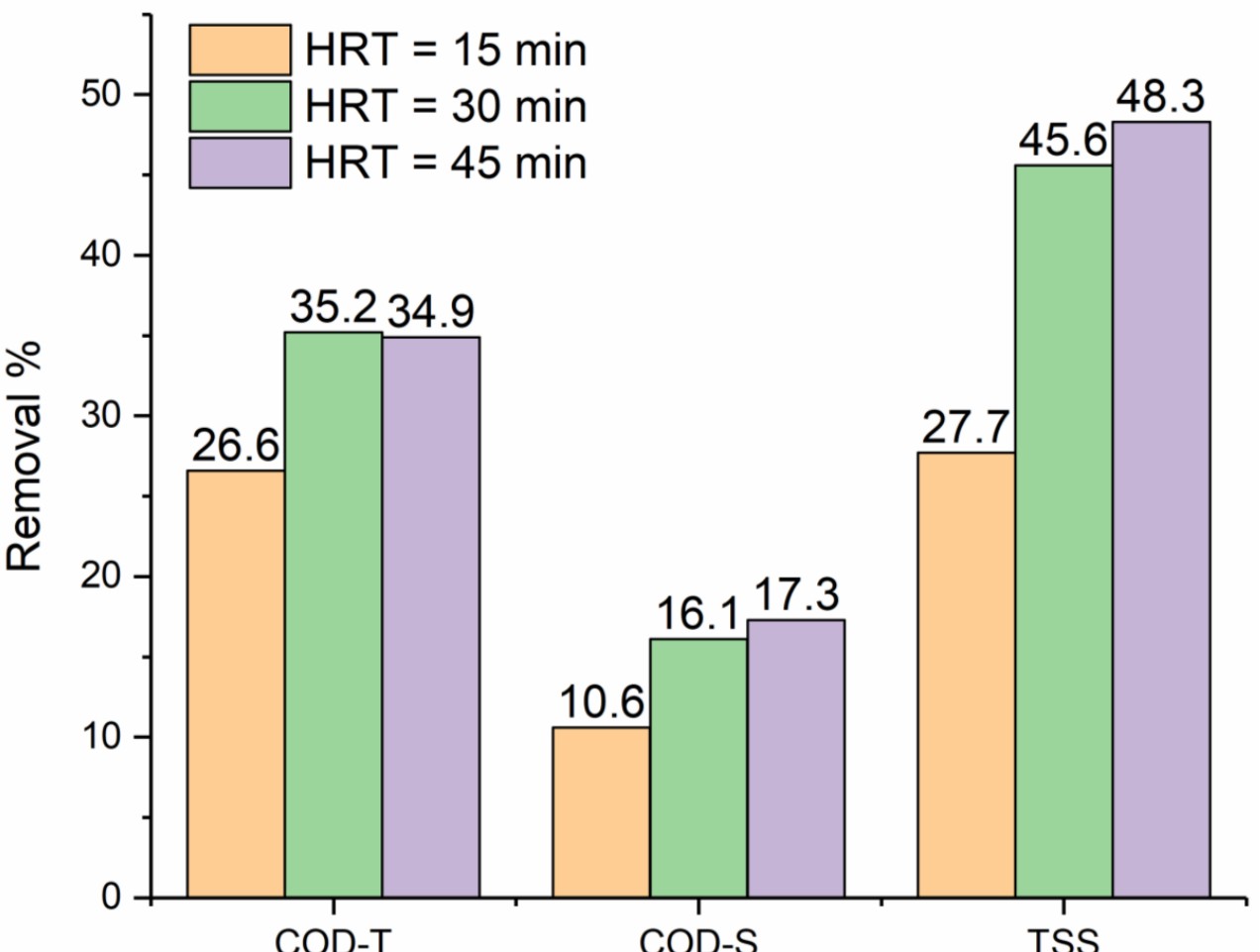

**Figure 3.** Effect of HRT on performance of pilot-scale BEPT process in terms of COD-T, COD-S and TSS removal.

### 3.5. Effect of Sludge Loading

Four different AS loadings, i.e., 0 mg/L, 60 mg/L, 80 mg/L and 100 mg/L, were selected based on the quantity of waste-activated sludge available from the SWRRF. It can be seen from Figure 4a that, without any AS addition, there was a slight removal of soluble COD (approximately 3%), while COD-T and TSS removals were 23% and 46%, which is comparable to PT. Moreover, at a low AS loading, i.e., 60 mg/L, COD-T and COD-S removals were enhanced by 12 and 13%, respectively, while there was no significant difference in TSS removal compared to the results without any AS addition. Increasing the AS loading to 80 mg/L (Figure 4c) further improved COD-T, COD-S and TSS removals. The highest AS loading of 100 mg/L showed further improvements, with average removals of 41% for COD-T, 22% for COD-S and 55% for TSS (Figure 4d).

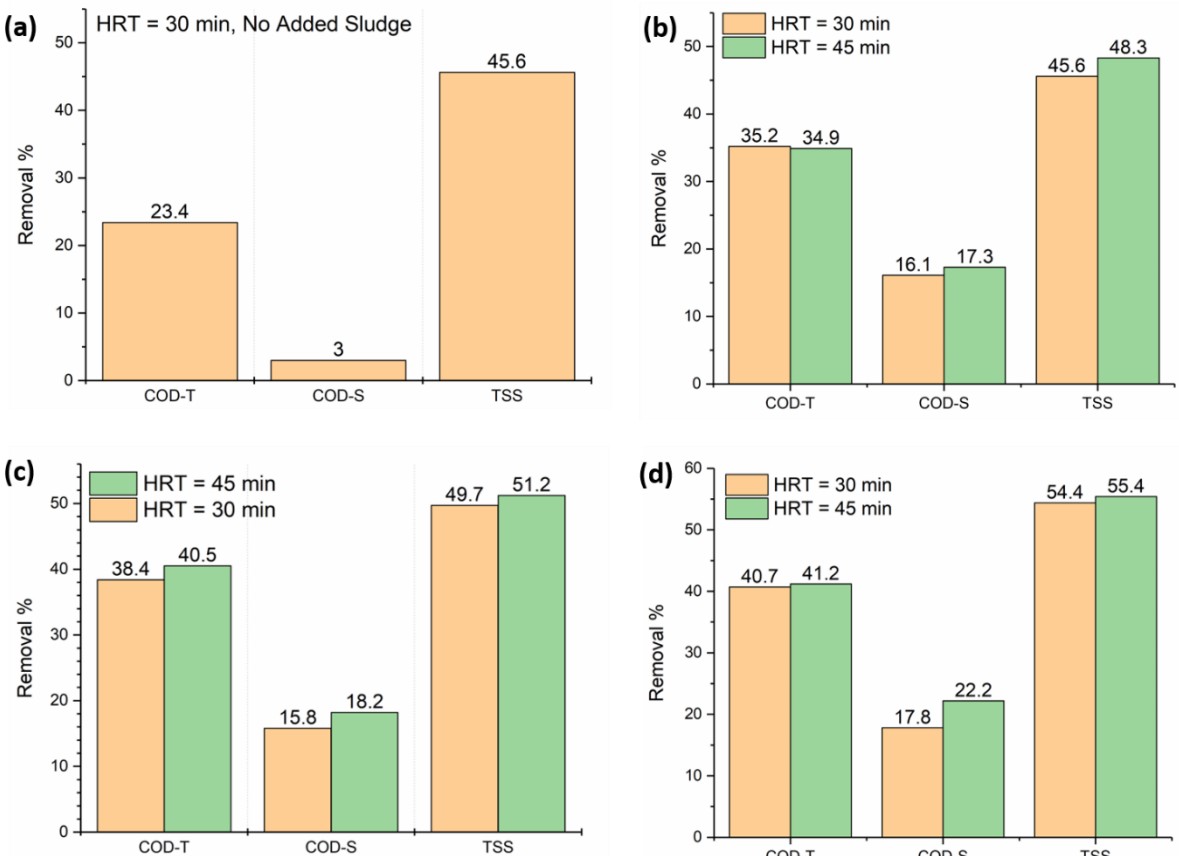

**Figure 4.** Effect of sludge loading on the performance of pilot-scale BEPT process with two HRTs; (**a**) 0 mg/L, (**b**) 60 mg/L, (**c**) 80 mg/L and (**d**) 100 mg/L sludge solids.

Overall, BEPT has higher capabilities compared to conventional PT. In BEPT-enhanced COD-T, COD-S and TSS removals, owing to quick HRT (30–45 min) and mild aeration (DO = 1 mg/L), enhanced removal efficiencies were found compared to PT. Moreover, the sludge produced from BEPT contained a high solid content (4.8 g/L), which may not require further thickening prior to introducing into an anaerobic digestor for biogas production. A graphical/schematic comparison between the BEPT and PT is shown in Figure 5.

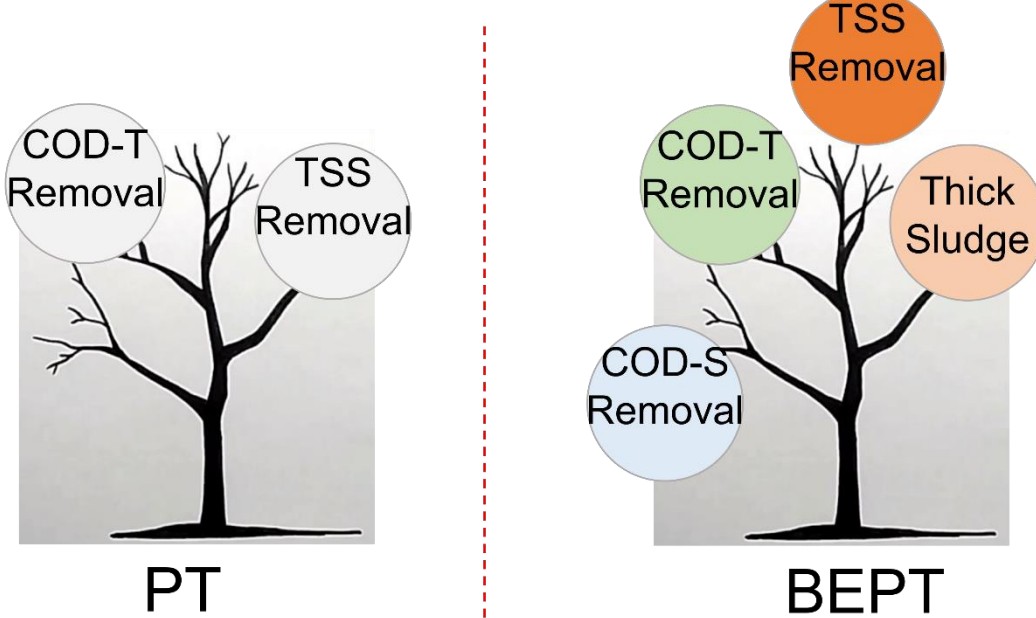

**Figure 5.** Schematic comparison between conventional PT and BEPT processes in primary treatment of wastewater.

## 4. Conclusions

Wastewater treatment plants are moving towards a paradigm shift to become resource recovery facilities. BEPT was successfully demonstrated without the addition of any flocculants or coagulants to capture organics from raw wastewater. DO, AS loading, and HRT were the most important parameters to optimize the performance of the BEPT process. It was found that, at a DO of 1 mg/L, an AS loading of 80 mg/L and an HRT of 30 min, BEPT could remove 38% of the COD-T and 16% of the COD-S on average. Overall, the studied configuration demonstrated promising results for carbon capture from actual wastewater without the addition of any chemicals.

**Supplementary Materials:** The following supporting information can be downloaded at: https://www.mdpi.com/article/10.3390/w14233825/s1, Figure S1: Equipment for bench scale testing of BEPT process. Figure S2: Generation of bubbles with tap water in pilot DAF contact zone. Figure S3: Performance of bench scale BEPT process.

**Author Contributions:** Conceptualization, M.K. and P.N.; methodology; software; validation, M.R.A. and K.C.; formal analysis; investigation; resources, M.K. and P.N.; data curation; writing—original draft preparation, M.R.A.; writing—review and editing, M.K. and K.C.; visualization; supervision; project administration; funding acquisition, M.K. All authors have read and agreed to the published version of the manuscript.

**Funding:** This research was funded by Water Corporation.

**Data Availability Statement:** Not applicable.

**Acknowledgments:** The financial contribution by Water Corporation for this work is appreciated.

**Conflicts of Interest:** The authors declare no conflict of interest.

## Abbreviations

| | |
|---|---|
| AS | activated sludge |
| BEPT | biologically enhanced primary treatment |
| CEPT | chemically enhanced primary treatment |
| COD | chemical oxygen demand |

| COD-S | soluble chemical oxygen demand |
| COD-T | total chemical oxygen demand |
| DAF | dissolved air flotation |
| DO | dissolved oxygen |
| HRA | high rate-activated sludge |
| HLR | hydraulic loading rate |
| HRT | hydraulic retention time |
| MBR | membrane bioreactor |
| PT | primary treatment |
| RAS | return activated sludge |
| TSS | total suspended solids |
| WWTPs | wastewater treatment plants |
| WRRF | Water Resource Recovery Facility |

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
