# Peer review of "Chemical-Free Biologically Enhanced Primary Treatment of Raw Wastewater for Improved Capture Carbon"

_water, doi:10.3390/w14233825_

Round 1

Reviewer 1 Report

The paper is about primary treatment of raw wastewater to capture carbon. Experimental work done by the authors and the result is very pleasant. The article is well written and informative; there was an appropriate research discussion with the tests. Therefore, it is recommended to publish in "Processes".

Other weaknesses to be corrected:

enter keywords, keywords should be in alphabetical order.

Author Response

Please find response in word file

Reviewer 2 Report

The authors conducted both bench-scale and pilot-scale studies on the modified DAF technique, with several parameter variations on influent flow rate, HRT, and sludge loading rate. The experiments were well planned, and the results are straight forward. However, more background of DAF and BEPT should be introduced. More details about the bench test setup and pilot study setup should be added. A thorough discussion is expected as well based on the testing results. Please also attach the supplementary materials as I didn’t find it in the system.

Specific comments are below.

Line 89: Please spell out the term “DAF”.

Line 90: What is Captivator technology?

Line 94: “bench top study” should be “bench scale study”?

Line 95-96: “Promising results are reported based on total and soluble COD-S and TSS.” This statement is vague. Please either specify how promising the results were or considering combining this statement into the previous sentence. Grammarly, it should be “ promising results have been reported…:.

Line 46-88: These two paragraphs are lengthy and unrelated to the topic in my opinion. I don’t see any experiments about energy recovery in solids treatment in the manuscript. I’d suggest the authors should consider delete them, or make them concisely. Instead, more background introduction should be presented regarding the DAF, and the biologically enhanced primary treatment, as I didn’t find any information in the Introduction. In reality, the BEPT or CEPT is not a common concept or popular technique compared to the advanced secondary/tertiary treatment techniques. Therefore, I’d like to see the comments from the authors on the advantages of primary treatment modification over secondary or tertiary treatment improvements if there is any.

Section 2.1:

- Please use past tenses to describe the experimental set-up.

- Is mixing tank the same as contact tank? Please mark the contact tank instead of mixing tank to avoid confusions on Figure 1.

- In Figure 1, was the pump submerged in the contact tank? Can you add the water level on both mixing tank and flotation tank? Is the sludge tank a real tank filled with the sludge?

-In Figure 1, please mark the flow direction on the effluent using an arrow.

In Figure 1, what is the green box with dashed line representing? Please clarify.

- Line 103: What is “pilot rig”? I feel that it’s due to my lack of knowledge on DAF. Please explain and define it.

Section 2.2:

-I am totally confused with the mixing tank and contact tank. If they are the same, please use a consistent name. If not, please identify them clearly.

-Line 124: What was the flow rate of sludge addition? How often did the authors refill the sludge tank? I saw the authors conducted the experiment with various flow rate. The variation of parameters should also be mentioned at a previous section.

-Line 129: Manually removed by a skimmer?

-Line 131: Where was the water sampling?

-The influent flow rate needs to be specified. Is the HRT of 1.5 hr in Line 130 for the entire DAF? It sounds like the HRT was for the scum (floating) storage tank.

Section 3.2:

-For the title, and other places throughout the manuscript, please use “benchtop” or “bench-top”.

-Line 149: Please be careful with the language and use past tenses to describe the experiment. It should be “no significant difference was observed”. ‘In COD removal efficiency” should be “in COD removal”. Please check the writings throughout the manuscript.

-Line 152: The definition of TSS as total suspended solids should be placed in Line 86.

Section 3.3:

-What’s the difference between the bench-scale and pilot-scale tests? It seems like the authors used the same set-up (DAF system) for both? More information and explanation should be added in the previous introduction and methods sections.

-Line 174: Please define PST.

-Table 1: Please add notes to define “COD-T” and “COD-S”.

-Line 175-177: Did the authors have any data to support this conclusion? I can’t find a result of solids production.

-Figure 2: It would be great to plot the data from PST for comparison.

Section 3.4:

-The discussions were focused on the nutrient removal. For example, in Line 192, it should be “a 10% increase in COD-T removal”. Please correct the entire section.

-Figure 3: It should be “effect of hydraulic retention time (HRT)…” in the caption. Please also clarify the time of this results, for example, at Day 200.

Section 3.5:

-Line 208: Please spell out ML.

-Line 209: the AS loading units need to be clarified. 60 mg/L as TSS? Was it the final solids concentration or the concentration of the sludge tank? In Line 141, the influent TSS concentration was 350-550 mg/L for the system. Again, the operation of DAF system needs to be documented in detail.

-Figure 4: Please add the sludge loading variations on the top of each sub-figure.

Section 3.6: The figure seems to be not necessary to me as it is not that informative. More discussions should be added in a thorough way. The authors should summarize the results from this study, comparing with previous studies, and propose an optimal operating condition based on the bench study and pilot study. A simple and general conclusion will not be considered as a Discussion.

A minor comment: I think using “total COD” and “soluble COD” would be more suitable throughout the manuscript, instead of defining them as “COD-T” and “COD-S”.

Author Response

Please find response in word file.

Reviewer 3 Report

This manuscript titled “Chemicals-Free Biologically Enhanced Primary Treatment of Raw Wastewater to Capture Carbon: A Case Study Towards Efficient Wastewater Treatment Plants.” describes the efficiency of biologically enhanced primary treatment of wastewater by using a high rate activated (HRAS) dissolved air flotation (DAF) system for real wastewater. I believe this report will add to scientific knowledge and practical aspects of wastewater treatment, however, I have few major concerns about this manuscript which should be addressed before this manuscript may be accepted for publication.

1.     Abstract needs to be rewritten so as to capture the main idea of this paper. As it is written, it is very generic and cannot provide the information needed for the reader to comprehend this study and its objectives. The abstract should avoid generic terms and present the actual work that was performed (objective of the work, methodology applied, and case study results).

2.     Authors should mention the composition and source of “real wastewater” in abstract section as it is not conveying the complete information what type and source of real wastewater is?

3.     Mention relevant keywords under heading “keywords.”

4.     Introduction section needs to be rewritten as by just mentioning” Although, WWTPs have been in use for more than a century…. (Page 2, line 46) does not convey proper message to the readers. Authors must mention different sources and origins of wastewater such as domestic wastewater, industrial wastewater, agriculture wastewater, etc. that were treated through WWTPs previously. In this context, authors may used some of the recent studies. E.g.,  Water Environment Research, 2022 (https://doi.org/10.1002/wer.1685), Water 2021, 13(22) https://doi.org/10.3390/w13223210; RSC Advances, 2016 (http://doi.org/10.1039/C5RA21076C).

5.     Page 2, line 96, “Herein, bench top study and pilot scale testing were carried using real municipal wastewater at Subiaco wastewater treatment facilities in Western Australia.” Authors must mention the composition and source of wastewater 9either domestic, industrial, agriculture etc…)

6.     Page 3, caption of Figure 1 should be rephrased. Authors must mention whether it is Schematic representation of experimental setup for HRAS-DAF system or BEPT system?

7.     Page 3, heading 2.1 should be rephrased as “Operational setup of BEPT through HRAS-DAF system”.

8.     Page 4, line 140 should be rephrased as “The raw wastewater was collected and tested directly from the wastewater treatment plant”.

9.     Authors must mention all the abbreviations in the introduction section so that they can easily use these abbreviations throughout the manuscript.

10.  The quality of figures needs to be promoted as the information in figures cannot be easily understood.

Round 2

Reviewer 2 Report

1. Please add marks indicating water level in the Figure 1 - contact tank and flotation tank.

2. Regarding the comments about water sampling location, I did not find the updates as the authors replied. Please add this information in the Methods section.

3. In one of the authors' replies, it was said that the retention time is for sludge instead of hydraulic retention time. If like that, using term of HRT is incorrect. It must be revised and defined clearly.

See below as the original reply from the authors.

-The influent flow rate needs to be specified. Is the HRT of 1.5 hr in Line 130 for the entire DAF? It sounds like the HRT was for the scum (floating) storage tank.

Thanks for the inquiry! This retention time only for retained sludge in floatation tank irrespective of hydraulic retention time.

4. In general, all abbreviations should be defined and spelled out when they appear the first time in the Introduction, even already being defined in the abstract. The list of abbreviations is not necessary if the authors follow above suggestions.

Author Response

1. Please add marks indicating water level in the Figure 1 - contact tank and flotation tank.

2. Regarding the comments about water sampling location, I did not find the updates as the authors replied. Please add this information in the Methods section.

Response:

The figure has been updated in revised manuscript, thank you!

  1. In one of the authors' replies, it was said that the retention time is for sludge instead of hydraulic retention time. If like that, using term of HRT is incorrect. It must be revised and defined clearly.

See below as the original reply from the authors.

-The influent flow rate needs to be specified. Is the HRT of 1.5 hr in Line 130 for the entire DAF? It sounds like the HRT was for the scum (floating) storage tank.

Thanks for the inquiry! This retention time only for retained sludge in floatation tank irrespective of hydraulic retention time.

Response: Please note, changes are made in revised manuscript as "The sludge float from the DAF unit was removed by a skimmer, and collected in a non-aerated tank with an approximate retention time of 1.5 h. " 

There is no mention of HRT in this sentence/discussion. 

4. In general, all abbreviations should be defined and spelled out when they appear the first time in the Introduction, even already being defined in the abstract. The list of abbreviations is not necessary if the authors follow above suggestions.

Response:

Thank you, all the abbbreviations are defined in the manuscript. 

Reviewer 3 Report

accepted in its present form.

Author Response

accepted in its present form.

Response:

Thank you!